# Design, Synthesis, and Cytotoxicity Assessment of [^64^Cu]Cu-NOTA-Terpyridine Platinum Conjugate: A Novel Chemoradiotherapeutic Agent with Flexible Linker

**DOI:** 10.3390/nano11092154

**Published:** 2021-08-24

**Authors:** Meysam Khosravifarsani, Samia Ait-Mohand, Benoit Paquette, Léon Sanche, Brigitte Guérin

**Affiliations:** 1Department of Nuclear Medicine and Radiobiology, Faculty of Medicine and Health Sciences, Université de Sherbrooke, Sherbrooke, QC J1H 5N4, Canada; Meysam.Khosravifarsani@USherbrooke.ca (M.K.); Samia.Ait-Mohand@USherbrooke.ca (S.A.-M.); Benoit.Paquette@USherbrooke.ca (B.P.); Leon.Sanche@USherbrooke.ca (L.S.); 2Sherbrooke Molecular Imaging Center (CIMS), CRCHUS, 3001, 12e Avenue Nord, Sherbrooke, QC J1H 5N4, Canada

**Keywords:** copper-64, chemoradiotherapeutic agent, cytotoxic activity, platinum-based compounds, terpyridine platinum, low energy electrons

## Abstract

Maximum benefits of chemoradiation therapy with platinum-based compounds are expected if the radiation and the drug are localized simultaneously in cancer cells. To optimize this concomitant effect, we developed the novel chemoradiotherapeutic agent [^64^Cu]Cu-NOTA-C3-TP by conjugating, via a short flexible alkyl chain spacer (C3), a terpyridine platinum (TP) moiety to a NOTA chelator complexed with copper-64 (^64^Cu). The decay of ^64^Cu produces numerous low-energy electrons, enabling the ^64^Cu-conjugate to deliver radiation energy close to TP, which intercalates into G-quadruplex DNA. Accordingly, the in vitro internalization kinetic and the cytotoxic activity of [^64^Cu]Cu-NOTA-C3-TP and its derivatives were investigated with colorectal cancer (HCT116) and normal human fibroblast (GM05757) cells. Radiolabeling by ^64^Cu results in a >55,000-fold increase of cytotoxic potential relative to [^Nat^Cu]Cu-NOTA-C3-TP at 72 h post administration, indicating a large additive effect between ^64^Cu and the TP drug. The internalization and nucleus accumulation of [^64^Cu]Cu-NOTA-C3-TP in the HCT116 cells were, respectively, 3.1 and 6.0 times higher than that for GM05757 normal human fibroblasts, which is supportive of the higher efficiency of the [^64^Cu]Cu-NOTA-C3-TP for HCT116 cancer cells. This work presents the first proof-of-concept study showing the potential use of the [^64^Cu]Cu-NOTA-C3-TP conjugate as a targeted chemoradiotherapeutic agent to treat colorectal cancer.

## 1. Introduction

Platinum-based drugs are used as first-line treatment for about half of all cancers, such as colorectal and those of the lungs, ovaries, testes, blood, head, and neck [1]. Many analogs of platinum (Pt) have been evaluated, but only cisplatin, oxaliplatin, and carboplatin are commonly used in clinic [1], although nedaplatin, heptaplatin, and lobaplatin have been limitedly authorized in a few countries [2]. The cytotoxicity of Pt-based drugs is due to their binding with DNA and the formation of DNA adducts, including intrastrand and interstrand crosslinks and monofunctional adducts [3]. These adducts disturb DNA conformation by destabilization of the double helix, which interferes with replication and the mitotic process, and ultimately induces cell death. Although their effectiveness is recognized, serious side reactions including nephrotoxicity, neurotoxicity, and ototoxicity can be induced, which prevent maximizing their antitumor effects by injecting higher doses [4]. In addition, resistance to Pt-based chemotherapy may be acquired or intrinsic [5]. For example, human colorectal cancer (CRC) cells may be intrinsically resistant to cisplatin and become so when treated with oxaliplatin [6,7,8]. With the aim to reduce side effects and overcome resistance, several families of platinum square planar complexes such as terpyridine platinum (TP) were synthesized. TP intercalates with a high affinity toward tertiary structures formed by guanine-rich motifs on DNA known as G-quadruplex [9,10,11,12]. These DNA structures play an important role in cell senescence and cell viability, making them potential therapeutic targets. Thus, a square planar complex of platinum such as (2,2′:6′,2″-terpyridine) platinum was used to induce cell senescence and cell death by increasing the stability of G-quadruplex [9,13,14].

Existing Pt drugs, especially cisplatin, are frequently used in combination with radiotherapy [15,16]. The enhanced efficiency of this combined treatment relative to chemo- and radiotherapy delivered separately has long been thought to be caused by the negative impact of Pt-drug adducts on the repair of radiation-induced damage [17]. On the other hand, recent investigations suggest that the presence of Pt-drug adducts in DNA acts at another level by increasing the initial yields of radiation-induced damage [18,19]. This enhancement appears to be relevant to the interactions of low-energy secondary electrons (LEEs) with energies <30 eV and their short penetration length (<10 nm) that are generated in large numbers during short times by the interaction of ionizing radiation with matter [20,21].

Considerable information has been compiled on the mechanisms of action of LEEs with DNA [18,19,22]. These fundamental studies have been used to predict that chemoradiation treatments should be more efficient when a high concentration of Pt-drugs binds to cancer cell DNA, which suggests delivering the radiation when their maximum binding is observed. This assumption has been validated in cell and animal models of CRC [6,23,24] and glioblastoma [25,26] for various Pt-derived chemotherapeutic agents in combination with external beam radiotherapy. Moreover, these studies suggest that DNA damage (and hence biological effectiveness) could be further maximized if LEEs are generated preferentially in large numbers close to Pt adducts within cellular DNA. For instance, it was observed that in unmodified DNA, electrons with the minimum energy of 5 eV are required to induce a double-strand break (DSB), whereas an individual near-zero eV electron could cause the same damage in platinum-modified DNA [27]. We hypothesized that those conditions could be optimized if a Pt drug is combined with a radioisotope, generating short-range LEEs.

Recently, we have designed an NOTA−TP-based conjugate with a planar and rigid linker containing two methylene groups attached to a central phenyl ring [28]. We have shown that this novel chemoradiotherapeutic agent exhibits a strong nanoscale supra-additive and selective cytotoxic effect when combined with copper-64 (^64^Cu). Considering that the length and flexibility of the linker may have an impact on the cytotoxic potency of this new class of TP-based conjugates, therefore, in the present study, we have synthesized a new conjugate with a TP moiety linked via a short flexible alkyl chain spacer (C3) to NOTA, which is a chelator that tightly binds ^64^Cu (Figure 1) [29,30]. This radioisotope (T_1/2_  =  12.7 h; EC, [43.1%], β^+^, 0.653 MeV [17.8%]; β^−^, 0.579 MeV [38.4%]) has decay schemes appropriate for positron emission tomography (PET) imaging via β^+^ emission and cancer therapy due to its decay by Auger electrons and β^−^ particles emission [31]. As described in the literature, the therapeutic effect is expected to come mainly from two Auger electrons with energies ranging from 840 eV to 6.5 KeV and a penetration distance ranging from 0.05 to 1.5 µm in soft tissue [32]. These primary electrons generate large amounts of LEEs having high linear energy transfer (LET), ranging from 2 to 25 keV/µm, and a relative biological effectiveness (RBE) that is 20 times greater than X-rays under in vitro conditions [19].

In the present study, the radiotherapeutic potential of [^64^Cu]Cu-NOTA-C3-TP was assessed for the first time in vitro in the human colorectal tumor cells HCT116 and in the normal fibroblast cell line GM05757.

## 2. Materials and Methods

### 2.1. Materials

All solvents and reagents were acquired from commercial suppliers and were used without further purification. NO2A*t*Bu was purchased from Chematech (Dijon France). The ^64^Ni target (99.52%) was purchased from Isoflex USA (San Francisco, CA, USA). Hydrochloric acid (99.999%), ammonium acetate (99.999%), sodium acetate (99.999%), trace metal basis, and sodium hydroxide pellets (≥98.0%) were obtained from Sigma-Aldrich (Saint-Louis, MO, USA). Acetonitrile (HPLC grade, 99.9%) and high-purity water (Optima LC/MS, ultra-high-performance liquid chromatography ultraviolet grade, 0.03 mm filtered) were purchased from Fisher Scientific (Ottawa, ON, Canada). ^1^H and ^13^C NMR spectra were recorded in deuterated solvents on a Brucker Ascend 400 NMR instrument. The NMR spectra are expressed on the *δ* scale and were referenced to residual solvent peaks and/or internal tetramethylsilane. All coupling constants (*J*) are in Hertz. The peak multiplicities are described as follows: s (singlet), d (doublet), t (triplet), q (quartet), quin (quintet), m (multiplet), and br (broad). Mass spectra were recorded on an API 3000 LC/MS/MS (Applied Biosystems/MDS SCIEX, Concord, ON, Canada), on a Waters/Alliance HT 2795 equipped with a Waters 2996 PDA and a Waters Micromass ZQ detector API 2000, and on ESI-Q-Tof (MAXIS). High-resolution mass was carried out through electrospray ionization using an on a Triple TOF 5600, ABSciex mass spectrometer. Instant thin-layer chromatography paper (ITLC-SA) was acquired from Agilent Technology (Santa Clara, CA, USA). All glassware was cleaned with chromic sulfuric acid (Fisher Scientific). The labeling efficiency of [^64^Cu]Cu-NOTA-C3-TP was assessed using ITLC-SG. The radio-TLC plates were scanned using an Instant Imager scanner Bioscan, DC, USA). Radioactivity measurements were done in an ionization chamber (CRC-25PET; Capintec) on the ^64^Cu setting to control process efficiency. [^64^Cu]CuCl_2_ was produced from enriched ^64^Ni targets using either a ACSI TR-19 or a TR24 cyclotron by the ^64^Ni(p,n)^64^Cu reaction using an enriched ^64^Ni target electroplated on a rhodium disc. [^64^Cu]CuCl_2_ was transformed in [^64^Cu]Cu(OAc)_2_ by dissolving the [^64^Cu]CuCl_2_ in ammonium acetate (0.1 M; pH 5.5), as described previously [33]. All solutions were made from distilled deionized water (Thermofisher Scientific, Barnstead Genepure).

**Compound 1:** In an inert atmosphere, 1,3-dibromopropane (1 mL, 4.92 mmol) dissolved in 20 mL of dry acetonitrile and K_2_CO_3_ (0.16 g, 1.17 mmol) were stirred at room temperature for 15 min followed by 5 min at 70 °C. Then, NO2A*t*Bu (0.20 g, 0.56 mmol) was added in small portion, and the mixture was stirred at 70 °C for another 72 h. The insoluble inorganic salt was filtered off, and the filtrate was concentrated in vacuo. Then, 50 mL of hexane/diethyl ether (1/1) was added to the residue, and the mixture was sonicated for 15 min. A yellow amorphous solid was obtained in 78% yield. ^1^H NMR (400 MHz, CDCl_3_): *δ*_H_ 4.52 (m, 6H), 3.30 (s, 4H), 2.92 (m, 4H), 2.78 (m, 3H), 2.52 (s, 5H), 1.42 (S, 18H). ^13^C NMR (CDCl_3_): *δ*_C_ 170.6, 82.0, 57.9, 53.7, 52.9, 52.9, 52.4, 49.7, 49.6, 49.6, 41.9, 29.8, 28.2, 28.2, 27.7, 27.6. ESI-MS Calcd for C_21_H_40_BrN_3_O_4_ [M]^+^: 478.47, Found [M + H]^+^: 480.18, HRMS (ESI-Q-Tof, *m*/*z*): Calcd: 478.2275, Found: 478.2274.

**Compound 2:** A mixture of 2,6-bis(2-pyridyl) 4(1*H*) pyridone (0.06 g, 0.22 mmol) and potassium carbonate (0.12 g, 0.83 mmol) in 15 mL of acetone was refluxed for 15 min. A solution of **1** (0.10 g, 0.21 mmol) dissolved in 5 mL of acetone was added dropwise to the flask, and the reaction mixture was heated under reflux for 48 h. Then, the reaction was “worked up” by removing the inorganic salts and adding water to the filtrate. The product was extracted with diethyl ether (20 mL × 2), and the combined organic extracts were washed with water, 5% sodium hydroxide, and water, and then dried with Na_2_SO_4_. After filtration, the solvent was removed under reduced pressure to yield a pale pink solid (43%). ^1^H NMR (400 MHz, CDCl_3_): *δ*_H_ 8.82 (dd, 2H, ^1^*J*_HH_ = 0.85, ^2^*J*_HH_ = 1.80), *δ*_H_ 8.69 (d, 2H, *J*_HH_ = 8.10), 8.07 (s, 2H), 7.98 (td, 2H, ^1^*J*_HH_ = 1.70, ^2^*J*_HH_ = 7.77), 7.45 (ddd, H, ^1^*J*_HH_ = 1.10, ^2^*J*_HH_ = 8.70, ^3^*J*_HH_ = 8.1), 4.41 (t, 2H, *J*_HH_ = 5.8), 3.51–3.62 (m, 6H), 3.42(s, 4H), 3.10–3.25 (m, 4H), 2.83 (s, 4H), 2.42 (m, 2H), 1.45 (s, 18H). ^13^C NMR (CDCl_3_): *δ*_C_ 171.1, 167.6, 162.0, 161.6, 156.1, 154.5, 148.7, 138.9, 125.1, 122.7, 118.4, 115.5, 108.7, 82.4, 66.2, 58.6, 58.6, 53.9, 52.7, 52.7, 28.5, 24.7. ESI-MS Calcd for C_36_H_50_N_6_O_5_ [M]^+^: 646.83, Found [M + H]^+^: 648.19, HRMS (ESI-Q-Tof, *m*/*z*): Calcd: 648.2895, Found: 649.2689.

**Compounds 3** and **4:** In a round-bottom flask, compound **2** (0.20 g, 0.31 mmol) was dissolved in 5 mL of methanol and then added to a solution of Pt(COD)Cl_2_ (0.12 mg, 0.32 mmol) dissolved in the same solvent. The resulting red solution was heated at 55 °C for 24 h. Then, the solution was cooled, and an aqueous solution of NaPF_6_ was added to precipitate out a light-yellow orange solid, which was centrifuged, washed with water many times, and dried. The product **3** was obtained as a very hygroscopic yellow solid (72%). ESI-MS Calcd for C_36_H_50_ClN_6_O_5_Pt [M]^+^: 876.37, Found [M + H]^+^: 877.65. This compound was directly dissolved in a minimum of acetone and 15 mL of hydrogen chloride solution 4.0 M in dioxane was added. The mixture was stirred at room temperature for 4–5 h. A very hygroscopic yellow orange solid was obtained as a NaPF_6_ salt (as described above) with 80% yield. ^1^H NMR (400 MHz, DMSO-d_6_): *δ*_H_ 11.74 (s, 2H, COOH), 8.90 (m, 2H), 8.70 (m, 2H), 8.52 m, 2H), 8.32 (s, 2H), 7.95 (m, 2H), 4.47 (s, 2H), 3.75 (m, 4H), 3.29 (m, 6H), 2.58–3.16 (m, 7H), 2.31 (m, 2H); ^13^C NMR (DMSO-d_6_): *δ*_C_ 171.7, 168.9, 158.4, 158.1, 158.1, 155.3, 151.0, 150.9, 142.5, 131.6, 129.2, 128.7, 126.1, 126.1, 110.9, 58.6, 55.7, 52.2, 49.9, 39.3, 39.0, 28.9. ESI-MS Calcd for C_28_H_34_ClN_6_O_5_Pt [M]^+^: 764.19, Found [M + H]^+^: 764.44, HRMS (ESI-Q-Tof, *m*/*z*): Calcd: 764.1924, Found: 764.1903.

**[^Nat^Cu]Cu-NOTA-C3-TP:** Preparation of [^Nat^Cu]Cu-NOTA-C3-TP conjugate followed the same procedure described for the first generation of the complex [28]. Ultimately, [^Nat^Cu]Cu-NOTA-C3-TP compound was isolated as a pale green PF_6_ salt (80%). ESI-MS Calcd for C_28_H_32_ClCuN_6_O_5_Pt [M]^+^: 826.68, Found [M + H]^+^: 827.72, HRMS (ESI-Q-Tof, *m*/*z*): Calcd: 826.1063, Found: 826.1032.

**[^64^Cu]Cu-NOTA-C3-TP:** The preparation of [^64^Cu]Cu-NOTA-C3-TP was performed by incubating 2 and 250 µM of compound **4** dissolved in 10% of DMSO in saline with 100–400 MBq of [^64^Cu]Cu(OAc)_2_ within 20–30 min at room temperature in 0.1 M ammonium acetate buffer pH 7.2. The radio chemical yield (RCY > 99%) was assessed by radio-TLC -and eluted on C18 plates using sodium citrate 0.1 M, pH = 5.5.

### 2.2. Stability Studies

Plasma stability of the compound was assessed by incubating [^64^Cu]Cu-NOTA-C3-TP in mouse plasma for 72 h at room temperature. Radio-TLC on C18 plates conducted on 0.1 M sodium citrate buffer at pH 5.5, as a developing solvent, was used to detect free ^64^Cu. [^64^Cu]Cu-NOTA-C3-TP and free [^64^Cu]Cu(OAc)_2_ were employed as control standards.

### 2.3. Cell Culture

Human CRC cells HCT116 obtained from ATCC were cultured in Eagle’s minimal essential medium (EMEM), while Dulbecco’s modified Eagle medium (DMEM) was used for the normal fibroblast cell line GM05757 purchased from Coriell Institute. These culture media were supplemented with 10% FBS, 26.2 mM sodium bicarbonate, 2 mM L-glutamine, and a mix of penicillin (100 UI/mL) and streptomycin (100 μg/mL). Cells were incubated at 37 °C in a humidified environment with 5% CO_2_.

### 2.4. Cytotoxicity Assay

Cytotoxic activity was measured with the PrestoBlue method, which consists of a Resazurin-based metabolic assay. Briefly, HCT116 cells (2 × 10^4^ cells per well) and GM05757 cells (5 × 10^3^ cells per well) were seeded in a 96-well plate and incubated for 24 h. Then, the cells were treated with the Pt drugs at concentrations ranging from 0 to 500 µM for a subsequent 24, 48, and 72 h. Cisplatin and oxaliplatin (diluted in 0.9% saline), and NOTA-C3-TP conjugates (diluted in DMSO) solutions were prepared just before use in EMEM or DMEM. The final concentration of DMSO was less than 1%. Then, the media were removed, and the cells were washed with warm PBS and exposed with PrestoBlue reagent for an incubation time of 20 min. Changes in cell viability were recorded by fluorescence (λ_ex_ = 570 nm; λ_em_ = 610 nm). The median effective concentration (EC_50_) values were calculated with dose–response curves, which were plotted as the concentration of Pt drug (x-axis) against fluorescence values (y-axis). The fluorescence values measured for blank groups were subtracted from those obtained from the treated and untreated cells.

### 2.5. Cellular Uptake, Internalization, and Efflux Assays in HCT116 Cells

HCT116 cells (2 × 10^5^ per well) were seeded in 12-well plates for 48 h in EMEM medium completed with 10% FBS at 37 °C until 80% confluence. Cells were washed twice with PBS, and then [^64^Cu]Cu-NOTA-C3-TP (0.1 MBq, 50 µL) in EMEM completed medium was added and incubated from 15 min to 72 h to measure cellular uptake and internalization. At each time point, the radioactive media was removed, and cells were washed thrice with cold PBS. To measure the net cellular uptake or internalized fraction, surface-bound Pt drugs were removed by treating with 50 mM glycine in PBS pH 2.2 for 10 min, which was followed by a second cold PBS wash. Then, cells were harvested through trypsinization and counted with a hemocytometer. The radioactivity was measured in a gamma counter (HIDEX Automatic Gamma Counter 2014). The results were expressed as a percentage of added dose (AD) retained per 10^6^ cells (%AD/10^6^ cells). Parallel experiments were carried out at 4 °C to assess internalization inhibition and surface-bound fraction. For efflux studies, HCT116 cells were incubated for 1 h with 0.1 MBq of [^64^Cu]Cu-NOTA-C3-TP. Then, the cells were washed with PBS and fresh culture medium was added. After 0 to 72 h, the medium was removed, and the cells were washed thrice with PBS. Finally, the cells were harvested, and radioactivity was counted with the gamma counter. The results were expressed as percentage of total activity retained by 10^6^ cells relative to baseline at 0 min.

For efflux studies, we followed the standard protocol of Fournier et al. [33]. Briefly, HCT116 cancer cells were incubated for 1 h with 0.1 MBq of [^64^Cu]Cu-NOTA-C3-TP. The activity retained by the cells was measured from 0 to 72 h post administration.

### 2.6. Subcellular Localization of [^64^Cu]Cu-NOTA-TP in HCT116 and GM05757 Cells

For extraction of HCT116 and GM05757 cell nucleus and cytoplasm, we followed a previously reported procedure [28,31]. After treatment of HCT116 cells with 1MBq of [^64^Cu]Cu-NOTA-C3-TP for 24, 48 and 72 h, CSK buffer (0.5% Triton X-100, 300 mM sucrose, 100 mM NaCl, 1 mM ethylenediaminetetraacetic acid (EDTA), 2 mM MgCl_2_, and 10 mM HEPES, pH 6.8) were employed for nuclear extraction. After 5 min centrifugation, supernatant was separated, and the retained activity in nuclear plates was counted by a gamma counter (HIDEX Automatic Gamma Counter 2014).

### 2.7. Data Analysis

All experiments were performed in triplicate, and the results were expressed as mean values ± S.D. A *p*-value of less than 0.05 was considered to be significant (two-tailed independent Student’s *t*-test). Calculations were performed with GraphPad Prism 7.0 (GraphPad Software, Inc., La Jolla, CA, USA).

## 3. Results

### 3.1. Synthesis and Characterization of [^64^Cu]Cu-NOTA-C3-TP

The design of our NOTA-C3-TP conjugate with a flexible spacer was inspired by a library of ligands developed by Stafford et al. showing that cyclen-TP-based bi-metallic complexes (with Cu^2+^, Pt^2+^, and Zn^2+^) had higher affinity toward G-quadruplex DNA as compared to their TP-based mono-metallic counterparts [13]. Considering these promising results, we propose to optimize the design of the TP conjugate by using a NOTA chelator instead of a cyclen for a stronger Cu^2+^ complexation with a radionuclide such as ^64^Cu, which shows great potential for use in radiotherapy and imaging [31,32].

The synthetic strategy used for the preparation of NOTA-C3-TP **4** involved a dialkylation/platinum insertion/deprotection sequence (Figure 1). As illustrated, the conjugate **4** was prepared in a multi-step process starting with commercially available NO2A*t*Bu (di-*tert*-butyl 2,2′-(1,4,7-triazacyclononane-1,4-diyl) diacetate, 1,3-dibromopropane, and 2,6-bis(2-pyridyl)-4(^1^H)-pyridone, with an overall yield of 20%. The first step is an alkylation of the bis-protected (NO2A*t*Bu) with 1,3-dibromopropane and potassium carbonate as a base to give 78% of compound **1** after 72 h of reflux in acetonitrile. Then, this compound was coupled with 2,6-bis(2-pyridyl)-4(^1^H)-pyridone in basic conditions at reflux in acetone to yield 43% of compound **2** isolated as a white pink solid. This NOTA-terpyridine conjugate was clearly identified by ^1^H and ^13^C NMR spectroscopy as well as by mass spectrometry. The reaction of NOTA-C3-TP with Pt(COD)Cl_2_ gives an orange solid, which was washed with water and diethyl ether, centrifuged, and dried to isolate **3** as a pure PF_6_^−^ salt with a 72% yield. In the final step, the *tert*-butyl ester groups were removed by HCl solution 4 M in dioxane to isolate **4** as a pure PF_6_^−^ salt with 80% yield. This complex was fully characterized by ^1^H and ^13^C NMR as well as by high-resolution mass spectrometry. We should mention that we could not isolated pure platinum complexes as chloride salts, so an additional counter-anion exchange step was required, so the chloride ion was exchanged for PF_6_^−^.

The [^Nat^Cu]Cu-NOTA-C3-TP conjugate was prepared by treating the compound with Cu(OAc)_2_ in 0.1 M ammonium acetate buffer, pH 5.5 for 60 min at room temperature. The copper complex was obtained as a pale green PF_6_ salt (80%). It is worth mentioning that the conjugate is not well soluble in water; 10% DMSO in saline is necessary to solubilize the [^Nat^Cu]Cu-NOTA-C3-TP. The labeling of NOTA-C3-TP with [^64^Cu]Cu(OAc)_2_ was completed within 20–30 min at pH 7.2 at room temperature with a non-decay corrected yield of 99%. After labeling, the molar activity of [^64^Cu]Cu-NOTA-C3-TP was ranged from 0.84 to 4 MBq/nmol, with radiochemical purity greater than 95% as determined by radio-TLC.

### 3.2. Stability of [^64^Cu]Cu-NOTA-C3-TP

[^64^Cu]Cu-NOTA-TP was stable up to 72 h in mouse plasma. No traces of free ^64^Cu or NOTA-TP fragments were detected by radio-iTLC over this time. Moreover, at the physiological pH 7.2, the amount of Pt released was not seen by mass spectroscopy after 72 h, suggesting a potential good stability before cellular internalization.

### 3.3. Cytotoxic Effects of [^64^Cu]Cu-NOTA-C3-TP

In order to fully underline the cumulative effect of [^64^Cu]Cu-NOTA-C3-TP, the cytotoxicity induced by NOTA-C3-TP and [^Nat^Cu]Cu-NOTA-C3-TP on the colorectal cancer cells HCT116 and normal fibroblasts GM05757 was first determined. The clinical standards cisplatin and oxaliplatin were included as references; see Table 1.

The cytotoxic activity of [^Nat^Cu]Cu-NOTA-C3-TP is maximal after 24 h incubation in HCT116 cells and significantly progressed with the incubation time for GM05757 cells (Table 1). The absence of ^Nat^Cu had a major effect on the activity of the conjugate. Indeed, NOTA-C3-TP was 2 to 14 times more cytotoxic (lower EC_50_ values) than [^Nat^Cu]Cu-NOTA-C3-TP. One should note that without Cu^2+^, the two carboxylic groups on NOTA are on the carboxylate form, and the conjugate has two negative charges at physiological pH. Nevertheless, the NOTA-C3-TP and [^Nat^Cu]Cu-NOTA-C3-TP conjugates were generally more toxic for the CRC cells HCT116 compared for the normal fibroblasts GM05757, suggesting a selective toxicity for this cancer cell line. They were also significantly much less cytotoxic than cisplatin (*p* < 0.02) after a short incubation time of 24 h. After a longer incubation of 72 h, the cytotoxicity of NOTA-C3-TP has increased to reach the EC_50_ values measured with the cisplatin and oxaliplatin, while [^Nat^Cu]Cu-NOTA-C3-TP remained much less toxic. The cytotoxic effect of cisplatin has reached a plateau after 24 h incubation, while the efficacy of oxaliplatin continued to improve with incubation time.

At low apparent molar activity (AMA, 0.8 to 4.0 MBq/nmol, entry 3), [^64^Cu]Cu-NOTA-C3-TP was slightly less cytotoxic than cisplatin, but it was 12, 5, and 3.4 times more active on HCT116 cells than NOTA-C3-TP, [^Nat^Cu]Cu-NOTA-C3-TP, and oxaliplatin respectively at 24 h post incubation (Table 1). The cytotoxic activity of [^64^Cu]Cu-NOTA-C3-TP increased over time, indicating 5–66-fold more activity on HCT116 cells relative to [^Nat^Cu]Cu-NOTA-C3-TP and NOTA-C3-TP at 48 and 72 h post administration. These results indicate that even at low AMA for [^64^Cu]Cu-NOTA-C3-TP, a significant improvement of its antitumor activity as compared to NOTA-C3-TP and [^Nat^Cu]Cu-NOTA-C3-TP (*p* < 0.001) was obtained. Interestingly, the cytotoxic activity of [^64^Cu]Cu-NOTA-C3-TP on GM05757 human fibroblast cells (EC_50_ > 200µM) was at least 3.4-fold lower than that of HCT116 cells. Parallel experiments were carried out with [^64^Cu]Cu(OAc)_2_ as control, and the results were reported previously [28]. Accordingly, the measured EC_50_ value to induce a 50% reduction in cell viability was 5.6 ± 0.5 in the CRC cells HCT116 at 24 h. This value is about 28 times higher than that for [^64^Cu]Cu-NOTA-TP at an EC_50_ concentration (0.2 MBq), suggesting that ^64^Cu alone is much less active than [^64^Cu]Cu-NOTA-TP toward HCT116 cancer cells.

By increasing AMA of [^64^Cu]Cu-NOTA-C3-TP to ≈120 MBq/nmol (entry 4), the antitumor activity of the ^64^Cu-conjugate was strikingly improved, resulting in EC_50_ values ranging from 17 ± 4 to 6 ± 2 nM at 24 to 72h post administration for the HCT116 cells. This represented an improvement by almost 17,000-, >40,0000-, and 55,000-fold with respect to [^Nat^Cu]Cu-NOTA-C3-TP at 24 h, 48 h, and 72 h, respectively (Table 1, entries 2 and 4). The cytotoxic activity of [^64^Cu]Cu-NOTA-C3-TP at high AMA is far superior to that of cisplatin and oxaliplatin at all time points tested. More importantly, in spite of this striking improvement in cytotoxicity of the [^64^Cu]Cu-NOTA-C3-TP at high AMA, it represented 2–4 times higher efficiency toward HCT116 cells relative to GM05757 normal fibroblasts. One should note that the highest selectivity index was measured at an earlier time point (24 h), when ^64^Cu deposits the highest dose rate of radioactivity in the cells.

The viability of the HCT116 cells in the presence of [^64^Cu]Cu-NOTA-C3-TP at low AMA was decreased to 22% and 17% when 5 and 8 MBq were respectively used during the incubation (Figure 2). The cytotoxic activity of [^64^Cu]Cu-NOTA-C3-TP on the HCT116 cells was reduced by adding NOTA-C3-TP (red and green lines). At a concentration of 300 µM of NOTA-C3-TP, the contribution of low dose of ^64^Cu (5 MBq) on the cytotoxic activity of HCT116 was reduced, but it continued to contribute significantly at a higher dose of ^64^Cu (8 MBq) compared to NOTA-C3-TP (*p* < 0.02) and [^Nat^Cu]Cu-NOTA-C3-TP (*p* < 0.02). By contrast, the cytotoxicity of [^Nat^Cu]Cu-NOTA-C3-TP on the HCT116 cells increased as its concentration increased. As illustrated in Figure 2, the cytotoxic effect of NOTA-C3-TP was considerably lower than those of both [^Nat^Cu]Cu-NOTA-C3-TP and [^64^Cu]Cu-NOTA-C3-TP compounds at 24 h incubation time in HCT116 cells. We hypothesize that the addition of NOTA-C3-TP to [^64^Cu]Cu-NOTA-C3-TP (red and green lines) at concentrations higher than 300 µM will eventually reach a similar effect on cell viability than NOTA-C3-TP alone (black line). All these results suggest that at higher concentration, most of the NOTA-C3-TP derivatives compete with the [^64^Cu]Cu-NOTA-C3-TP for the same targets and contribute to reducing its cytotoxicity.

### 3.4. Cellular Uptake and Internalization of [^64^Cu]Cu-NOTA-C3-TP

The cell viability assays have shown that the ^Nat^Cu- and [^64^Cu]Cu-NOTA-C3-TP were more toxic against the CRC cells HCT116 than for the normal fibroblasts GM05757 (Table 1). To investigate the mechanism involved, the cellular uptake kinetics of [^64^Cu]Cu-NOTA-C3-TP were determined during 72 h at 37 °C in these two cell lines. The ^64^Cu-conjugate fraction weakly bound to the cell surface was removed with a mild acid buffer.

The cellular uptake kinetic of [^64^Cu]Cu-NOTA-C3-TP in the CRC cells HCT116 was faster than that measured with the fibroblasts GM05757. A maximum accumulation was measured after 24 h incubation, which was followed by a slow decrease (Figure 3). The percentages of internalization for this ^64^Cu-conjugate in the HCT116 cancer cells were 3.1, 1.8, and 2.1-fold higher at 24 h, 48 h, and 72 h post administration than those measured with the GM05757 fibroblasts (Figure 3). These results suggest that the highest efficiency of the [^64^Cu]Cu-NOTA-C3-TP for the HCT116 cancer cells was at least in part associated with a more efficient cellular accumulation. These results are consistent with a reduced uptake of TP complexes in GM05757 fibroblasts relative to cancer cells reported by Suntharalingam et al. [34].

Then, the kinetic of total cell-associated and internalized fraction of [^64^Cu]Cu-NOTA-C3-TP in the HCT116 cells was determined (Figure 4A). The ^64^Cu-conjugate was added to the HCT116 cells at 37 °C, and the total cell-associated activity and the internalized fraction were measured from 15 min to 48 h later. The results of total cell-associated fraction correspond to the radioactivity measured before the wash with the mild acid buffer. As seen in Figure 4A, the total cell-associated and internalization fraction of [^64^Cu]Cu-NOTA-C3-TP increased over time and reached a maximum value after 24 h of incubation. The internalized fraction increased strikingly from 0.04 ± 0.02% after 15 min incubation to its maximum value (18.7 ± 2.8%) at 24 h, which is consistent with our previous results for a chemoradiotherapeutic agent labeled with ^64^Cu [28]. This kinetic is also consistent with those observed with TP compounds as reported by Stafford et al. [13]. A similarly kinetic was found for the cell-associated fraction of the [^64^Cu]Cu-NOTA-C3-TP, but 2-fold higher values were measured. The maximum was measured at 24 h (40.0 ± 2.8%), after which it slowly decreased (Figure 4A).

The passive accumulation of the ^64^Cu-conjugate in the HCT116 cells was assessed by repeating the assays at 4 °C (Figure 4B). The [^64^Cu]Cu-NOTA-C3-TP has accumulated at a much slower rate and reached an uptake ~30-fold lower than that measured at 37 °C, which represented less than 1% of the [^64^Cu]Cu-NOTA-C3-TP incubated with these cells. These results suggest that passive accumulation has a weak role in the internalization of this TP conjugate.

### 3.5. Kinetic of [^64^Cu]Cu-NOTA-C3-TP Efflux

The efflux assay was done to determine whether [^64^Cu]Cu-NOTA-C3-TP was quickly cleared from HCT116 cells (Figure 5A). After an incubation of 1 h with the HCT116 cells, the ^64^Cu-conjugate was removed, and its kinetic of efflux was measured during 72 h. About 42% of TP conjugate was rapidly eliminated during the first 6 h, which was followed by a slow efflux rate, as more than 40% of [^64^Cu]Cu-NOTA-C3-TP was still retained by the HCT116 cells 72 h post administration. This kinetic of drug efflux corresponds to the initial rapid elimination of ^64^Cu-fraction associated to the cell surface, which was followed by the slow removal of [^64^Cu]Cu-NOTA-C3-TP that was internalized. Then, the assay was repeated by adding [^Nat^Cu]Cu-NOTA-C3-TP at 100 or 500 µM (Figure 5B). The rapid elimination of [^64^Cu]Cu-NOTA-C3-TP in the presence of an excess of non-radioactive conjugate suggests that its internalization was occurring via specific carriers.

### 3.6. Nuclear Localization of [^64^Cu]Cu-NOTA-C3-TP

The significant enhancement in the uptake of [^64^Cu]Cu-NOTA-C3-TP by HCT116 cancer cells relative to GM05757 fibroblasts could influence their subcellular distribution, particularly accumulation in the nucleus, as a potential target for the induction of DNA damage via low-range LEEs emitted by ^64^Cu. To verify this possibility, subcellular fractionation was carried out for both cell types.

Interestingly, the nucleus localization of [^64^Cu]Cu-NOTA-C3-TP at high AMA was measured to be 8.6 ± 3%, 13.4 ± 4%, and 12.3 ± 1% in HCT116 cells at 24 h, 48 h, and 72 h, respectively (Figure 6A), whereas it was 1.5 ± 0.2%, 3 ± 0.4%, and 3 ± 0.1% in GM05757 fibroblasts at similar time points (Figure 6B). These results indicate 5.7-, 4.4-, and 4.1-fold reduction in the GM05757 fibroblast cell nucleus accumulation of [^64^Cu]Cu-NOTA-C3-TP as compared to that for HCT116 cancer cells at 24 h, 48 h, and 72 h post administration, respectively. In the case of cytoplasmic distribution, a range of percentages between 76 ± 7.3%, 61 ± 11%, and 61 ± 8.3% in HCT116 cancer cells (Figure 6A), and 46 ± 6%, 53 ± 6.5%, and 58.5 ± 8.3% in fibroblasts were calculated at 24 h, 48 h, and 72 h, respectively (Figure 6B). The subcellular localization of [^64^Cu]Cu(OAc)_2_ in HCT116 cells was investigated as a control group, and the results were reported previously [28].

## 4. Discussion

Previous experiments have shown that the addition of external beam radiation improves by 11.4-fold the cytotoxicity of cisplatin when it reached its maximum accumulation in DNA of the colorectal cancer cells HCT116 [6]. However, the optimal time to irradiate in a clinical setting is difficult to achieve. The external radiation beam also delivers a uniform dose of radiation through a cancer cell. Therefore, the benefit of concomitant chemoradiotherapy could be more easily obtained if the radiation is concentrated on the DNA where the drug Pt is linked. To reach this goal, the radioisotope ^64^Cu was used for the labeling of a NOTA-C3-TP conjugate. TP complexes are known to intercalate into G-quadruplex [9,10,11,12], which is targeted to induce cell senescence and cell death [9,13,14]. The decay of ^64^Cu produces two Auger electrons with a penetration distance ranging from 0.05 to 1.5 µm in soft tissue [32], and they generate large amounts of short-range LEEs [19]. Therefore, this strategy has the potential to concentrate the energy deposited by the radiation close to the platinum atom, in a sensitive DNA target, thus providing chemoradiation therapy at the molecular level.

It appears that the complexation of a Cu^2+^ atom to NOTA-C3-TP significantly reduces its cytotoxicity. While the EC_50_ of [^Nat^Cu]Cu-NOTA-C3-TP remained stable over time, that for the Cu-free NOTA-C3-TP decreases such that the EC_50_ value for HCT116 cells reaches 24 ± 1 µM, which corresponds to a similar cytotoxicity to cisplatin after 72 h of incubation (23 ± 3 µM). Based on our results, NOTA-C3-TP and its ^64/Nat^Cu-conjugates are internalized via the same receptor(s). At this time, there is no clear explanation for the increased cytotoxicity of NOTA−C3-TP over ^nat^Cu- NOTA−C3-TP on cancer cells.

The cytotoxicity of [^Nat^Cu]Cu-NOTA-C3-TP for the HCT116 cells was very low, as it shows an EC_50_ of 298 ± 2 µM, which is 10-fold less cytotoxic than cisplatin after 24 h of incubation. However, the addition of ^64^Cu had a major impact. At a concentration of 100 µM, only 0.1% of the NOTA-C3-TP was labeled with ^64^Cu, which resulted in a 5-fold increase of cytotoxicity. The accumulation of [^64^Cu]Cu-NOTA-C3-TP in HCT116 cells has mainly occurred through an active transporter, which still has to be identified. Although the EC_50_ value of cisplatin in μM concentration is inferior to that of [^64^Cu]Cu-NOTA-C3-TP at low AMA, the cellular accumulation of the latter was rapid, with a maximum measured after 24 h incubation, which corresponded to 72 ng of Pt/10^6^ HCT116 cells; i.e., it was 14.4 times larger than cisplatin (5 ng Pt/10^6^ HCT116 cells) [35]. The higher ability of [^64^Cu]Cu-NOTA-C3-TP to accumulate in HCT116 cells can be seen as a major advantage compared to cisplatin, as a much more important amount of the ^64^Cu radioisotope could be uptaken by the cancer cells and then intercalated into the G-quadruplex DNA.

Interestingly, we found that the AMA of [^64^Cu]Cu-NOTA-C3-TP has a large impact on its cytotoxicity. The EC_50_ values of [^64^Cu]Cu-NOTA-C3-TP at high AMA are in the low nanomolar range at all time points. Under these conditions, this radiotherapeutic agent is far more potent than all non-radioactive platinum compounds tested. These results suggest that increasing the AMA of ^64^Cu and the toxicity of Cu-NOTA-C3-TP could significantly amplify the concomitant chemoradiotherapeutic effect of this new generation of Pt drugs. These results also support the notion that a local delivery of Auger electrons with a radioisotope such as ^64^Cu close to platinum atom could be more efficient than combining Pt drug with an external radiation beam, which is deposited uniformly in cancer cells. The improvement of cytotoxicity observed at elevated AMA can be related to a lower competition with the non-radioactive NOTA-C3-TP, resulting in a higher cellular and nuclear internalization of the ^64^Cu-TP complex and maximizing intercalation into the G-quadruplex DNA and DNA damage arising from the radiotherapeutic properties of ^64^Cu. This hypothesis will have to be validated by measuring the radiation-absorbed dose of [^64^Cu]Cu-NOTA-C3-TP to the HCT116 nucleus in a future study. In this regard, the nucleus percentage of [^64^Cu]Cu(OAc)_2_ was compared to that of [^64^Cu]Cu-NOTA-C3-TP [28]. Interestingly, we measured 2.8-, 4.9-, and 3.7-fold increases in the HCT116 nucleus uptake of [^64^Cu]Cu-NOTA-C3-TP relative to [^64^Cu]Cu(OAc)_2_ at 24, 48, and 72 h, respectively. This result provides further evidence for striking improvement in the cytotoxicity of [^64^Cu]Cu-NOTA-C3-TP.

TP complexes interact non-covalently with G-quadruplex DNA and induce cell death via a different mechanism than cisplatin, which forms covalent adducts with DNA [9,10,11,12,34]. Cisplatin accumulation can occur through copper transporters such as the uptake transporter hCtr1 and the polyspecific organic cation transporter of hOCT1 [36]. Conversely, they can be pumped out through the efflux transporters ATP7A and ATP7B, suggesting a contribution of these transporters to the sensitivity of cells [37]. Therefore, it is expected that the resistance mechanisms of the two categories of Pt drugs would be different. Consequently, the Auger electrons emitter [^64^Cu]Cu-NOTA-C3-TP may provide an opportunity to circumvent the observed resistance to platinum drug in cancer cells [38].

The cytotoxicity of our NOTA-C3-TP conjugates and those of cisplatin and oxaliplatin were significantly higher against the HCT116 cells than against normal fibroblasts GM05757. Most significantly, the chemo-radiotherapeutic window of the [^64^Cu]Cu-NOTA-C3-TP at high AMA was considerably greater than that of non-radioactive NOTA-C3-TP conjugates or oxaliplatin (i.e., the clinical standard for CRC treatment) within the first 24 h of incubation, which is when the highest radiation dose deposition occurs. Several cellular and molecular mechanisms may account for the higher sensitivity of cancer cells to platinum drugs relative to normal human cells. These include the rapid cell division of cancer cells, an impaired ability to recognize and repair DNA damage that eventually leads to cell death by mitotic catastrophe or apoptosis, an overexpression of high mobility group (HMG) protein complexes in cancer cells, as well as the level and longevity of reactive oxygen species (ROS) generated in cancer cells after exposure to platinum drugs that could trigger a specific signaling transduction pathway and subsequently the activation of P38MAPK and JNK factors that mediate apoptosis [39].

Our results suggest that the highest efficiency of [^64^Cu]Cu-NOTA-C3-TP for the HCT116 cancer cell was at least in part associated with a more efficient cellular and nucleus accumulation than that obtained in the GM05757 fibroblasts. This finding is consistent with previous observations showing about a five-fold lower cellular uptake of another platinum (II)-terpyridine complex in the same GM05757 fibroblasts relative to human cancer cells [34]. Note that the epidermal growth factor receptor (EGFR), which is often overexpressed in cancer cells, has been recently reported as a potential target for platinum (II)-terpyridine compounds [40]. This molecular pathway could be a possible mechanism for higher internalization as well as the subcellular accumulation of [^64^Cu]Cu-NOTA-C3-TP in HCT116 cancer cells. This hypothesis needs to be validated in a future study.

Finally, although a strong supra-additive and selective cytotoxic effect was obtained when combining ^64^Cu with the first generation of NOTA-TP conjugate [28], the design of this chemoradiotherapeutic agent may not have been optimal. In the current study, we showed that the use of a short alkyl chain to link the NOTA chelator to the TP moiety resulted in a structure having considerable flexibility and adaptability, which was critical for its activity. Indeed, the ^64^Cu-conjugate with the short alkyl linker presented higher selectivity toward cancer cells (3.9 vs. 1.6) and the percentage of nucleus internalization (8.6% vs. 4.1%) at an early time point (24 h) compared with the [^64^Cu]Cu-NOTA-TP conjugate with a rigid linker [28]. These results suggest that [^64^Cu]Cu-NOTA-C3-TP could be a more effective chemoradiotherapeutic agent against CRC.

## 5. Conclusions

This proof-of-concept supports the possible use of the [^64^Cu]Cu-NOTA-C3-TP conjugate as a novel chemoradiotherapeutic agent for the treatment of primary tumors and potentially metastatic cancers. Our cellular results showed that [^64^Cu]Cu-NOTA-C3-TP accumulated rapidly and reached a higher level in the HCT116 cells than cisplatin, while its cellular efflux rate was slow. The new ^64^Cu-TP complex with a flexible linker has high specificity for the CRC HCT116 cells and is very cytotoxic at high AMA. Therefore, in vivo validation of [^64^Cu]Cu-NOTA-C3-TP at high AMA and investigation of potential intercalation-based mechanism represents the next steps to further evaluate the preclinical potential of this promising chemoradiotherapeutic agent to treat primary tumors and metastatic cancers.

## Data Availability

The data presented in this study are available in [this article].

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
