# Peer review of "Design, Synthesis, and Cytotoxicity Assessment of [64Cu]Cu-NOTA-Terpyridine Platinum Conjugate: A Novel Chemoradiotherapeutic Agent with Flexible Linker"

_nanomaterials, 2021, doi:10.3390/nano11092154_

Round 1

Reviewer 1 Report

Comment#1:

In paragraph: 2.2 Stability studies: indicate temperature of incubation.

Comment#2:

Concerning cytotoxicity studies, the compound NOTA-C3-TP appears to be more toxic than natCu-NOTA-C3-TP. This suggests that copper reduces the toxicity of the compound. It thus seems important to have a control group studying the cytotoxic effect of [64Cu]Cu(OAc)2 on HTC116 cells.

Comment#3:

For cellular uptake studies, a control group treated with [64Cu]Cu(OAc)2 is missed.

Comment#4

A direct comparison between NOTA−TP conjugates and NOTA−C3-TP in cytotoxicity studies using the same cell lines is essential to demonstrate the impact of the short flexible alkyl chain spacer (C3).

Author Response

Dear Reviewer,

Regards,

Brigitte 

Reviewer 2 Report

The paper describes a new therapeutic agent based on synergistic interaction of platinum based and radioactive moieties. The study is well planned and precisely performed.

I can recommend some minor changes to be made:

r.193-194 and r.206 Cell number should be indicated as cells per well and there is a substantial difference between 96- and 12-well plate in order to ensure a logarithmic growth of cultured cells.

Fig. 2. Standard deviations of the data points could be added to the curves.

The separation method of nuclear/cytoplasm fractions is only referred. A short description of the procedure and confirmation markers can be added. 

Author Response

Dear Reviewer,

Regards,

Brigitte 

Reviewer 3 Report

The manuscript entitled “Design, synthesis and cytotoxicity assessment of [64Cu]Cu-NOTA-terpyridine platinum conjugate: a novel chemoradiotherapeutic agent with flexible linker” describes the possible use of [64Cu]Cu-NOTA-C3-TP conjugate as a novel chemoradiotherapeutic agent for the treatment of primary tumors and potentially metastatic cancers. Cellular results showed that [64Cu]Cu-NOTA-C3-TP accumulats rapidly and reaches a higher level in the HCT116 cells than cisplatin, while its cellular efflux rate is slow. The new 64Cu-TP complex with a flexible linker has high specificity for the CRC HCT116 cells and is very cytotoxic at high AMA. In vivo validation of [64Cu]Cu-NOTA-C3-TP at high AMA and investigation of potential intercalation-based mechanism therefore represent the next steps to further evaluate the preclinical potential of this promising chemoradiotherapeutic agent to treat primary tumors and metastatic cancers.

It suggests to use at least two human CRC cell lines to identify the effects.

Author Response

Dear Reviewer,

Regards,

Brigitte 

Round 2

Reviewer 1 Report

The changes made to the documents allow a better understanding of the added value of this work and provide the necessary clarifications for a better understanding of the article.